# World Model Implanting for Test-time Adaptation of Embodied Agents

**Minjong Yoo** [1]  **Jinwoo Jang** [1]  **Sihyung Yoon** [1]  **Honguk Woo** [1]

## Abstract

In embodied AI, a persistent challenge is enabling agents to robustly adapt to novel domains without requiring extensive data collection or retraining. To address this, we present a world model implanting framework (WorMI) that combines the reasoning capabilities of large language models (LLMs) with independently learned, domain-specific world models through test-time composition. By allowing seamless implantation and removal of the world models, the embodied agent's policy achieves and maintains cross-domain adaptability. In the WorMI framework, we employ a prototype-based world model retrieval approach, utilizing efficient trajectory-based abstract representation matching, to incorporate relevant models into test-time composition. We also develop a world-wise compound attention method that not only integrates the knowledge from the retrieved world models but also aligns their intermediate representations with the reasoning model's representation within the agent's policy. This framework design effectively fuses domain-specific knowledge from multiple world models, ensuring robust adaptation to unseen domains. We evaluate our WorMI on the Virtual-Home and ALFWorld benchmarks, demonstrating superior zero-shot and few-shot performance compared to several LLM-based approaches across a range of unseen domains. These results highlight the framework's potential for scalable, real-world deployment in embodied agent scenarios where adaptability and data efficiency are essential.

## 1. Introduction

In recent years, policies powered by large language models (LLMs) have demonstrated remarkable success in embodied

---
[1]Department of Computer Science and Engineering, Sungkyunkwan University. Correspondence to: Honguk Woo <hwoo@skku.edu>.

*Proceedings of the 42nd International Conference on Machine Learning*, Vancouver, Canada. PMLR 267, 2025. Copyright 2025 by the author(s).

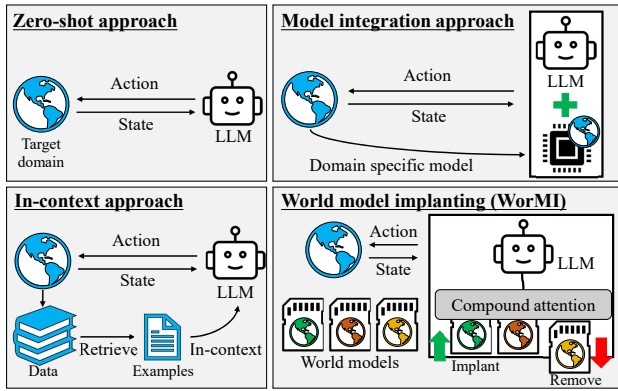

Figure 1: Concept of WorMI. For sequential decision-making policies of embodied agents, our WorMI framework, a **world model implanting** approach, is built upon an adaptive and composable policy structure that incorporates world-to-world integration and world-to-reasoning (world-to-LLM) alignment stages, enabling the flexible, test-time fusion of domain-specific knowledge. This dual-stage design not only combines the strengths of **model integration** (which embeds a domain-specific model as part of the policy) and **in-context adaptation** (which incorporates examples relevant to target domains) but also significantly enhances adaptability to unseen domains.

AI, a field focused on creating intelligent agents capable of making sequential decisions and interacting with the physical environment through tasks such as navigation and manipulation (Huang et al., 2022a;b; Yao et al., 2022; Wang et al., 2023). However, a significant challenge remains in enabling agents to adapt effectively to unseen domains without the need for extensive data collection or retraining. This adaptability is crucial for real-world applications, where environmental variations and diverse objectives often render rigid, domain-specific policies inadequate or ineffective.

Figure 1 illustrates different approaches used for policy learning in the field of LLM-based embodied agents, including in-context adaptation (Song et al., 2023) and model integration (Brohan et al., 2023; Hazra et al., 2024). The in-context adaptation approach retrieves relevant data from multiple domains to identify the most pertinent information for a given situation. While offering flexibility to some extent, it often suffers from inefficiencies due to the over-

head of searching, retrieving, and processing large volumes of information. On the other hand, the model integration approach combines two or more models with distinct properties, explicitly separating domain-specific aspects (e.g., affordance (Brohan et al., 2023), cost (Hazra et al., 2024)) from general knowledge. While this aims for efficient use of prior knowledge, it inherently lacks the flexibility to expand knowledge beyond its learned domains. Our direction enhances these approaches by incorporating the test-time model composition into embodied policy adaptation.

Specifically, we present a **world model implanting** framework, WorMI, which enables an embodied policy to adapt across diverse domains through test time, dual-stage composition of domain-specific world models. This process encompasses both world-to-world integration and world-to-reasoning alignment, where LLMs' reasoning capabilities are leveraged. By seamlessly implanting and removing world models upon varied target domains, the agent's policy tends to maintain grounded reasoning capabilities. It not only effectively leverages knowledge from multiple domains but also flexibly expands its knowledge, selectively utilizing only the most relevant information.

Our WorMI framework integrates two key methods into an adaptive, composable policy structure tailored for LLM-based embodied agents. (a) A **prototype-based world model retrieval** method selectively activates only a set of relevant world models. To determine relevance, each model's similarity to the current target domain is measured using object-wise state embeddings and clustering outcomes derived from trajectory-based prototypes. This ensures a more robust and interpretable adaptation process across diverse domains, particularly in zero-shot or few-shot scenarios. (b) A **world-wise compound attention** method effectively integrates the world models with the reasoning model by adaptively combining the pertinent knowledge from the retrieved model set. This facilitates effective and efficient policy adaptation during test-time execution. The interplay of these two methods enables the agent to dynamically compose and contextualize domain-specific knowledge in its policy, through coherent integration and alignment of world models and a reasoning model.

Through experiments with VirtualHome (Puig et al., 2018), and ALFWorld (Shridhar et al., 2021), we demonstrate that the WorMI framework achieves competitive performance in both effectiveness and efficiency compared to several state-of-the-art LLM-based embodied agents. For example, WorMI improves the average success rate in VirtualHome over SayCanPay (Hazra et al., 2024), a state-of-the-art LLM-based embodied agent, by 20.41% in zero-shot and 26.58% in few-shot scenarios. Our contributions are summarized as follows.

- We present the novel WorMI framework to enable cross-domain embodied policy adaptation at test time, exploring the dual-stage model compositionality: world-to-world knowledge integration and world-to-reasoning alignment.

- We implement the prototype-based retrieval method to efficiently provide a robust adaptation process over multiple domain-specific world models.

- We develop the world-wise compound attention method, where the hierarchical cross-attention between the world models and the reasoning model is meta-learned to enable the dynamic integration of pertinent domain-specific knowledge.

- We demonstrate the superiority of WorMI through experiments in zero-shot and few-shot scenarios on the VirtualHome and ALFWorld benchmarks, highlighting its robustness across varied domains as well as its data-efficient ability to handle unseen domains.

## 2. Related Work

**LLM-based Embodied Agent.** LLM-based embodied agents, whose tasks are often referred to as embodied instruction following, interact with the physical environment through activities such as object manipulation and navigation and require a high-level understanding of environmental dynamics and observations. Recent research has explored various techniques to enhance the reasoning and planning capabilities of these agents. They include implementing code-driven policies (Singh et al., 2023; Liang et al., 2023), generating reward functions (Yu et al., 2023; Adeniji et al., 2023; Kim et al.), and integrating LLMs with additional domain-specific models to harness the affordances or values of skills in the environment (Brohan et al., 2023; Hazra et al., 2024). Furthermore, in-context learning approaches, which incorporate previous demonstrations as part of the reasoning process, have also been introduced (Song et al., 2023).

While these approaches underscore the flexibility and robustness of LLMs for embodied instruction following, they often rely on external and disconnected integration of environmental data or additional models, thus limiting their cohesion. Furthermore, they rarely address the challenge of effectively integrating multiple domain-specific models to facilitate adaptation to unseen domains. In contrast, our work introduces a scalable approach that implants multiple world models into LLM reasoning, enabling a highly flexible and easily replaceable policy. This design facilitates strong generalization across diverse domains and rapid adaptation, achieved through attention-driven, coherent knowledge integration from carefully selected multiple models.

**Cross-domain Policy Adaptation.** To tackle the challenge

of handling diverse environmental features and tasks, researchers have investigated a range of domain generalization methods (Zhou et al., 2022). Representative approaches include meta-learning, which learns how to learn multiple tasks (Finn et al., 2017; Nichol, 2018; Andrychowicz et al., 2016; Ha et al., 2016), hierarchical learning, which structures knowledge across multiple levels by decomposing domain properties, and ensemble learning, which trains domain-specific neural networks for more robust performance. However, these methods often fall short of fully leveraging the specialized knowledge contained in domain-specific models. Our framework extends the benefits of these existing approaches, by implanting diverse, domain-specific world models into a single policy. This enables effective adaptation across a wide range of domains.

**Model Merging.** Researchers have explored a variety of model merging methods that bring multiple models together in collaboration, often achieving performance levels that exceed those of any single model. These approaches, encompassing model fusion (Wan et al.; Jiang et al., 2023), agentic workflows (Wu et al.; Hong et al.), and ensemble techniques (Shazeer et al., 2016), capitalize on the complementary strengths of individual component models to enhance overall performance. Recently, CALM (Bansal et al., 2024) introduced a model composition framework that connects two models via a cross-attention layer for tasks like translation or multi-language coding, allowing the two models to combine their respective representations. Unlike CALM, we focus on embodied agents that operate in the physical environment, requiring adaptive decision-making across diverse and unseen domains. To address this, our WorMI framework employs a compound attention mechanism that dynamically combines and aligns multiple world models within the agent's policy. This enables the agent to flexibly fuse relevant knowledge at test time, thus ensuring robust reasoning and optimal decision-making.

# 3. Approach

## 3.1. Overall Framework

A key challenge in LLM-based embodied agents is composing domain-specific knowledge into a general reasoning model, enabling agents to handle a wide range of unseen domains. We present the WorMI framework which enables an agent to dynamically retrieve and compose relevant world models at test time, fusing them into its LLM-based policy. This approach aims at ensuring robust zero-shot and few-shot adaptation across diverse domains in ever-changing environments while avoiding the need for retraining or fine-tuning large models.

In WorMI, we assume the availability of pre-trained world models $M_1, \ldots, M_N$ along with corresponding datasets

$\mathcal{D}_1, \ldots, \mathcal{D}_N$ from domains $D_1, \ldots, D_N$. A reasoning model $\pi_R$, built upon an LLM, provides general reasoning and decision-making capabilities. Given these individual models, we introduce a trainable composition module $C_\theta$ that selectively integrates only the relevant subset of pre-trained world models $\{M_1, \ldots, M_K\}$ with the fixed reasoning model $\pi_R$. Our hierarchical approach first fuses multiple world models into a unified representation, which then aligns with $\pi_R$ to form the implanted policy $\pi_\theta = C_\theta(\{M_1, ..., M_K\}, \pi_R)$.

As illustrated in Figure 2, to achieve this, our framework WorMI integrates two methods into an LLM-based policy structure. (a) **Prototype-based world model retrieval:** at test time, only the most relevant world models for the current target domain are retrieved and integrated. To achieve this, we assess similarity using trajectory-based prototypes, which are derived from embedding and clustering outcomes of object-wise states. This ensures that the selected world models closely align with the agent's current environment and task requirements. (b) **World-wise compound attention:** we train a world-wise compound attention module to integrate the intermediate representations of multiple world models and then align them with the reasoning model, enabling effective and efficient knowledge integration. Meta-learning on various subsets of world models makes the compound attention flexibly integrate domain-specific knowledge from any combination of world models, even when new world models are added. With this hierarchical attention mechanism, which aligns the integrated representations of domain-specific knowledge for reasoning, WorMI facilitates zero-shot and few-shot adaptation at test time.

## 3.2. Prototype-based World Model Retrieval

**World Models.** Each domain-specific world model $M_j$ is trained on its associated dataset $\mathcal{D}_j = \{(I, s_t, a_t, s_{t+1})\}$, where $I$ denotes the embodied task instructions, $s_t$ is the agent's state at time $t$, and $a_t$ is the executed action. Training comprises three auxiliary tasks: the dynamics task, predicting the next state $s_{t+1}$ given $(s_t, a_t)$; the action affordance task, identifying feasible actions $a_t$ from $s_t$; and the behavior cloning task, learning a policy $p(a_t \mid s_t, I)$. Collectively, these tasks equip each world model with domain-specific knowledge of transitions, affordances, and decision-making.

**Prototype-based Retrieval.** For a given state $s_t$, we retrieve a set of world models $\mathbf{M}_{\text{ret}}$ based on their relevance to $s_t$. This selection is made by measuring the embedding set distance between the set of embeddings $\mathcal{E}_j$ which encodes object-wise states in the environment for each dataset $\mathcal{D}_j$ and the set of embeddings $\mathcal{E}$ derived from $s_t$. Accordingly, the retrieved set $\mathbf{M}_{\text{ret}}$ can be formalized as

$$\mathbf{M}_{\text{ret}} = \left\{ M_j \ \middle| \ j \in \text{TopK}\Big( \{-\delta(\mathcal{E}_j, \mathcal{E})\}_{j=1}^N, \ K \Big) \right\} \quad (1)$$

**(a) Prototype-based world model retrieval**

**(b) World-wise compound attention**

Figure 2: Overall procedure of WorMI. (a) By the prototype-based world model retrieval, relevant world models are selected using trajectory-based prototypes derived from object-wise state embeddings, given target domains. (b) By the world-wise compound attention, multiple world models' representations are integrated and then aligned with the fixed reasoning model for effective knowledge fusion.

where $K$ is the number of selected world models and $\delta(\cdot, \cdot)$ is the Wasserstein distance. However, directly computing such a set distance incurs a high computational cost and tends to over-represent frequently occurring objects in the dataset. Therefore, we adopt a prototype-based similarity to reduce computational overhead, especially at test time while maintaining representational diversity.

To obtain such reliable prototypes, we construct the embedding set $\mathcal{E}_j$ in an object-wise manner, using the object detection model $\Phi_D$ that transforms input states to object-wise ones and the embedding model $\Phi_E$ that can be implemented using a language or vision embedding model. We have

$$\mathcal{E}_j = \{\Phi_E(o) \mid o \in \{o_1, ..., o_n\} = \Phi_D(s), s \in \mathcal{D}_j\} \quad (2)$$

where $o$ denotes object-wise states for a state $s$. We derive the prototypes by clustering $\mathcal{E}_j$ using the $k$-center method, identifying the top-$k$ embeddings that minimize the maximum distance from any point to its nearest center. This approach both retains crucial object embeddings and reduces the need for extensive distance computations over the entire dataset, formally described as

$$\mathbf{p}_j = \underset{C \subseteq \mathcal{E}_j, |C|=k}{\arg\min} \left[ \max_{x \in \mathcal{E}_j} \min_{c \in C} \|x - c\| \right]. \quad (3)$$

These $k$ centers serve as the prototypes $\mathbf{p}_j$ for each world model $M_j$. Using $\mathbf{p}_j$, which comprises only about 0.1% of the total embeddings in $\mathcal{E}_j$, and $\mathbf{p}$ from the current observation's embedding set $\mathcal{E}$, we replace $\delta(\mathcal{E}_j, \mathcal{E})$ with $\delta(\mathbf{p}_j, \mathbf{p})$ in (1). In our implementation with VirtualHome, the number of embeddings in $\mathcal{E}_j$ is 10,200, and $K$ is 15.

**Boundedness of Prototype-based Similarity.** Here, we show that the distance between prototype sets can serve

as a bounded proxy for the distance between the underlying datasets. Let $\rho$ be the minimal maximum distance, optimized by (3), such that each point in $\mathcal{E}_i$ is within $\rho$ of a prototype in $\mathbf{p}_j$. Then, applying the triangle inequality under the Wasserstein distance yields the following.

$$\delta(\mathbf{p}_i, \mathbf{p}_j) \le \delta(\mathcal{E}_i, \mathcal{E}_j) + 2\rho \quad (4)$$

### 3.3. World-wise Compound Attention

To integrate a set of world models with the reasoning model in a policy, we develop the world-wise compound attention method, which utilizes hierarchical cross-attention to effectively fuse domain-specific knowledge and align it with LLM reasoning at test time.

**Compound Attention.** The compound attention $C_\theta$ maps the $i^{th}$ layer outputs of world models $l_{M_1}, l_{M_2}, ..., l_{M_K}$ and $j^{th}$ layer output of the reasoning model $l_{\pi_R}$ to inputs of $(j+1)^{th}$ layer of the reasoning model.

$$l_{\pi_R} + C_\theta(\{l_{M_1}, l_{M_2}, ..., l_{M_K}\}, l_{\pi_R}) \quad (5)$$

As illustrated in Figure 3, the compound attention $C_\theta$ consists of a linear projection layer $L_\theta$, world-level cross-attention layer $Attn_\theta^W$, and reasoning-level cross-attention layer $Attn_\theta^R$. The linear projection layer matches the dimensions of the world model's intermediate-layer representation with those of the reasoning model. The world-level cross-attention layer integrates these intermediate representations through a weighted combination. The reasoning-level cross-attention layer then aligns the integrated representation with the reasoning model based on its queries.

**Linear Projection.** We first apply a linear projection $L_\theta$ to each $l_{M_j}$, matching its dimension to that of the reasoning

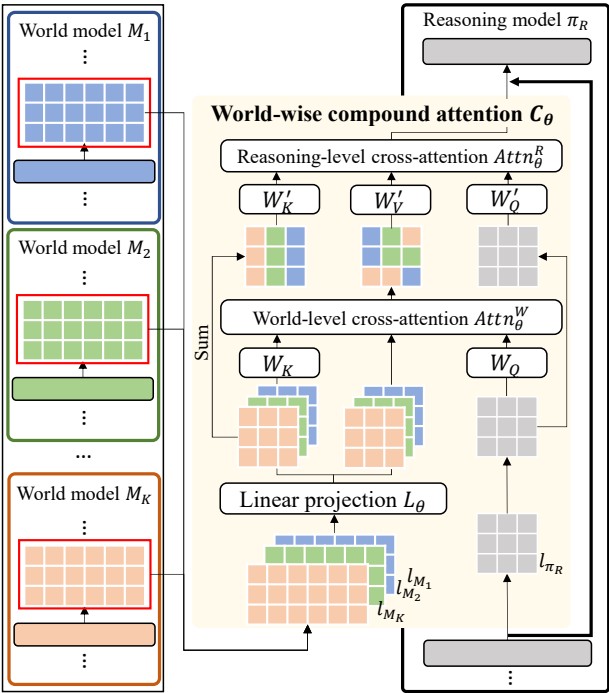

Figure 3: Overall structure of the compound attention. Compound attention allows for the test-time composition of multiple world models and the reasoning model. The world-level cross-attention first integrates the representation of multiple world models. The reasoning-level cross-attention then aligns the integrated representation to the reasoning model. Together, these hierarchical attention layers enable the policy to flexibly attend to relevant domain-specific knowledge and tightly couple it to reasoning.

model's embedding space.

$$\hat{l}_{M_j} = L_\theta(l_{M_j}), \quad j \in \{1, 2, \ldots, K\} \quad (6)$$

**World-level Cross-attention.** The world-level cross-attention $Attn_\theta^W$ quantifies the relative importance of each world model's representation, conditioned on the current hidden state of the reasoning model. The query of the world-level cross-attention is derived from the reasoning model's layer output $l_{\pi_R}$. We treat each output of world model $\hat{l}_{M_1}, \hat{l}_{M_2}, \ldots, \hat{l}_{M_K}$ as key-value pairs. Then, the query, key, and value of cross-attention $Attn_\theta^W$ are defined as

$$Q_W = W_Q \, l_{\pi_R}, \ K_W = W_K \left[\hat{l}_{M_1}; \ldots; \hat{l}_{M_K}\right],$$
$$V_W = \left[\hat{l}_{M_1}; \ldots; \hat{l}_{M_K}\right]. \quad (7)$$

Here, $W_Q, W_K$ are learned matrices in $\theta$. The output of $Attn_\theta^W$ produces an integrated representation, determining the relevance of each world model and assigning greater attention to the most pertinent ones for the given queries.

**Reasoning-level Cross-attention.** After obtaining the integrated representation of world models via the world-level cross-attention, we align it to the reasoning model. In this step, the query is derived from $\hat{l}_{\pi_R}$, the keys are the sum of the world models' outputs $\hat{l}_{M_1} + \ldots + \hat{l}_{M_K}$, and the values are the aggregated representation from the world-wise attention $Attn_\theta^W(Q_W, K_W, V_W)$. Then, the query, key, and value of cross-attention $Attn_\theta^R$ are defined as

$$Q_R = W_Q' l_{\pi_R}, \ K_R = W_K' \left[\hat{l}_{M_1} + \ldots + \hat{l}_{M_K}\right],$$
$$V_R = W_V' Attn_\theta^W(Q_W, K_W, V_W). \quad (8)$$

### 3.4. Meta-Learning for Model Implanting

To enable rapid adaptation to unseen domains, WorMI incorporates a meta-learning approach (Nichol, 2018) for its compound attention $C_\theta$, treating it as a parameter-efficient composer that dynamically aggregates and aligns domain-specific knowledge from multiple world models rather than merely linking existing models with the reasoning model.

**Learning Algorithm**. At each inner-loop update, the parameters $\theta_j$ are initialized by meta-parameters $\theta$. They are then adapted to such a subset of world models $\mathbf{M}_j = \{M_1, \ldots, M_m\}$ and datasets $\mathbf{D}_j = \mathcal{D}_1 \cup \ldots \cup \mathcal{D}_m$. This enables the compound attention $C_\theta$ to learn how to weigh and align features from each $\mathbf{M}_j$ for task planning. Formally, the loss function $\mathcal{L}$ is calculated as

$$\mathcal{L}(\theta_j, \mathcal{B}) = \sum_{(s,a) \in \mathcal{B}} -\log \pi_{\theta_j}(a|s) \quad (9)$$

for a batch $\mathcal{B}$ sampled from $\mathbf{D}_j$. In the outer-loop update, we combine the adapted parameters back into the meta-parameters by $\theta \leftarrow \theta + \beta \cdot \frac{1}{m} \sum_{j=1}^{m} (\theta_j - \theta)$ where $\beta$ is the learning rate for the outer-loop update. This process encourages $C_\theta$ to learn a general integration and alignment strategy that can be specialized to an unseen domain with only a few gradient steps. Consequently, compound attention acts as a composer, fusing domain-specific knowledge in the reasoning process and enabling rapid adaptation to both new domains and newly introduced world models.

## 4. Experiments

**Environments and Datasets.** We evaluate our approach in two embodied environments: VirtualHome (Puig et al., 2018), a 3D simulation environment for household task execution, and ALFWorld (Shridhar et al., 2021), a text-based environment for indoor task simulation through language interaction. We treat tasks and scenes as separate domains for adaptation. For VirtualHome, we collect 1,023 episodes, covering 78 tasks (16 seen, 62 unseen) across 20 distinct scenes (6 seen, 14 unseen), each featuring unique room layouts and objects. For ALFWorld, we collect 3,554 episodes

Table 1: Zero-shot performance in VirtualHome and ALFWorld. We use the 95% confidence interval, using 5 random seeds.

| Model | Seen Tasks & Seen Scenes | | Seen Tasks & Unseen Scenes | | Unseen Tasks & Unseen Scenes | |
|---|---|---|---|---|---|---|
| | SR ($\uparrow$) | PS ($\downarrow$) | SR ($\uparrow$) | PS ($\downarrow$) | SR ($\uparrow$) | PS ($\downarrow$) |
| **Evaluation in VirtualHome** | | | | | | |
| ZSP | 11.15%±0.65% | 29.02±0.08 | 8.95%±1.13% | 29.25±0.11 | 8.19%±0.33% | 29.36±0.56 |
| LLM-FT | 58.55%±2.18% | 17.37±0.34 | 53.42%±0.79% | 17.70±0.23 | 42.82%±0.76% | 20.79±0.17 |
| LLM-Planner | 35.67%±1.25% | 27.15±0.13 | 28.55%±0.42% | 27.04±0.12 | 21.45%±0.42% | 27.73±0.05 |
| SayCanPay | 69.88%±2.32% | 14.53±0.55 | 64.74%±0.87% | 15.64±0.18 | 45.71%±0.59% | 19.04±0.63 |
| WorMI | **85.78%±0.45%** | **10.76±0.19** | **80.26%±1.02%** | **12.42±0.09** | **66.12%±0.80%** | **15.17±0.08** |
| **Evaluation in ALFWorld** | | | | | | |
| ZSP | 2.30%±0.21% | 49.34±0.66 | 2.26%±0.09% | 49.04±0.95 | 2.13%±0.09% | 49.68±0.23 |
| LLM+FT | 45.72%±0.39% | 14.63±1.35 | 29.26%±0.72% | 38.49±2.87 | 29.40%±1.65% | 46.84±0.15 |
| LLM-Planner | 17.73%±0.61% | 32.09±2.90 | 18.63%±0.82% | 40.50±2.57 | 12.31%±0.80% | 46.33±1.68 |
| SayCanPay | 40.67%±1.24% | 18.37±1.93 | 34.10%±1.17% | 20.82±1.21 | 39.66%±1.43% | 23.86±1.90 |
| WorMI | **62.51%±1.65%** | **9.96±1.29** | **52.67%±1.39%** | **17.74±0.83** | **51.67%±2.23%** | **20.18±0.63** |

across diverse scenes. Following CL-ALFRED benchmark settings (Kim et al., 2024), the data is clustered into 4 scene types (3 seen, 1 unseen) and 6 task types (4 seen, 2 unseen).

**Evaluation Metrics.** We employ two evaluation metrics: Success Rate (SR) and Pending Steps (PS). SR measures the proportion of tasks completed in VirtualHome and sub-goals completed in ALFWorld. PS represents the average number of timesteps required to complete tasks, akin to cost-effectiveness in Hazra et al. (2024).

**Baselines.** For comparison, we evaluate four baselines that span diverse cross-domain adaptation strategies. **ZSP** (Huang et al., 2022a) is a zero-shot approach that applies a pre-trained model to new domains without additional adaptation. **LLM+FT (Fine-Tuned LLM)** performs adaptation through fine-tuning on limited domain-specific data. **LLM-Planner** (Song et al., 2023) employs in-context learning to generate and refine high-level plans based on examples. Lastly, **SayCanPay** (Hazra et al., 2024) is a state-of-the-art LLM-based model integrated approach that incorporates heuristic cost minimization into LLM reasoning and planning. These baselines collectively allow us to assess both the efficiency and robustness of our WorMI.

In our implementation, we use a fixed Llama-3.2-3B (AI@Meta, 2024) model for ZSP, LLM-Planner, the Say model in SayCanPay, and the reasoning model in WorMI. For LLM+FT, the Pay model in SayCanPay, and the world models in WorMI, we use a trainable Llama-3.2-1B model.

### 4.1. Main Results

**Zero-shot Adaptation.** We evaluate each method in a zero-shot scenario, relying solely on seen domain training data without any target domain data. As shown in Table 1, WorMI consistently outperforms all baselines across both seen and unseen domains, particularly demonstrating robust generalization to unseen domains (unseen tasks and

scenes). Specifically, WorMI outperforms the most competitive baseline SayCayPay in unseen tasks and scenes, achieving a 20.41% increase in SR and a 3.87 step reduction (20.32% improvement) in PS in VirtualHome, and a 12.01% increase in SR and a 3.68 reduction (18.23% improvement) in ALFWorld. These zero-shot results on robust generalization for unseen domains highlight the effectiveness and data efficiency of knowledge integration from multiple world models in WorMI. Moreover, the improvement in seen domains demonstrates the impact of the world-to-reasoning alignment which can enhance the agent's understanding of the environment.

**Few-shot Adaptation.** We further evaluate few-shot adaptation scenarios, where each target domain (unseen during training) provides only a small number of demonstrations at test time. This setup allows us to examine how effectively WorMI adapts with minimal additional data. Table 2 compares performance across different few-shot settings (1, 5, and 10 shots), illustrating how increasing the available demonstrations affects adaptation quality. As shown, WorMI surpasses all baselines, achieving on average a 26.58% gain in SR and 4.98 step reduction in PS in VirtualHome, and a 19.16% gain in SR and 7.36 step reduction in PS in ALFWorld, compared to SayCanPay. These results indicate that the lightweight compound attention module, trained via meta-learning, provides parameter-efficient updates while keeping the world models and the reasoning model frozen, thereby facilitating rapid, efficient adaptation even in minimal data conditions.

### 4.2. Analysis

We conduct several analyses on WorMI with unseen tasks and scenes in VirtualHome.

**World-level Attention Map.** In Figure 4, we visualize the attention weights of the world-level cross-attention used in the compound attention of WorMI. We employ three

Table 2: Few-shot performance in VirtualHome and ALFWorld. We use the 95% confidence interval, using 5 random seeds.

| Model | 1-Shot | | 5-Shot | | 10-Shot | | Average | |
|---|---|---|---|---|---|---|---|---|
| | SR (↑) | PS (↓) | SR (↑) | PS (↓) | SR (↑) | PS (↓) | SR (↑) | PS (↓) |
| **Evaluation in VirtualHome** | | | | | | | | |
| LLM-FT | 42.35%±0.85% | 20.46±0.15 | 47.22%±0.46% | 16.82±0.29 | 51.45%±1.06% | 16.82±0.39 | 47.01%±0.79% | 18.03±0.28 |
| LLM-Planner | 24.63%±0.34% | 26.19±0.31 | 29.80%±0.55% | 26.98±0.08 | 33.49%±0.44% | 26.60±0.16 | 29.31%±0.44% | 26.59±0.18 |
| SayCanPay | 46.75%±1.02% | 19.12±0.16 | 49.80%±1.11% | 15.52±0.11 | 56.24%±0.79% | 16.00±0.13 | 50.93%±0.97% | 16.88±0.13 |
| WorMI | **74.90±1.57%** | **12.18±0.31** | **78.04%±0.34%** | **11.85±0.08** | **79.61%±0.99%** | **11.68±0.13** | **77.51%±0.98%** | **11.90±0.17** |
| **Evaluation in ALFWorld** | | | | | | | | |
| LLM-FT | 26.78%±1.20% | 46.51±0.27 | 29.79%±1.36% | 46.21±0.37 | 33.57%±0.98% | 43.70±0.83 | 30.05%±1.18% | 45.47±0.49 |
| LLM-Planner | 18.28%±1.06% | 43.94±2.87 | 16.80%±1.46% | 40.50±3.14 | 17.76%±1.25% | 42.45±2.64 | 17.61%±1.26% | 42.30±2.88 |
| SayCanPay | 40.94%±1.34% | 21.58±1.14 | 36.70%±1.31% | 22.15±3.55 | 39.54%±1.48% | 24.74±3.48 | 39.06%±1.38% | 22.82±2.72 |
| WorMI | **51.50%±2.21%** | **21.46±3.96** | **58.69%±1.94%** | **13.14±1.00** | **64.46%±0.88%** | **11.79±0.81** | **58.22%±1.68%** | **15.46±1.92** |

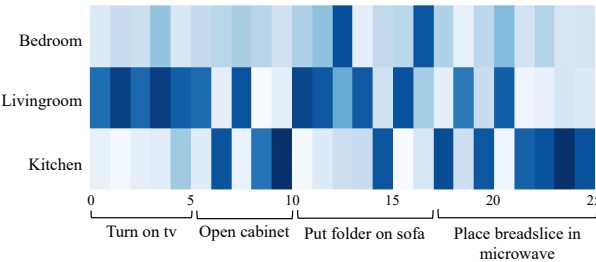

Figure 4: Visualization of world-level attention map

world models, each derived from different rooms (e.g., bedroom, livingroom, and kitchen), to facilitate adaptation to unseen domains. We observe that WorMI dynamically shifts its focus among the three world models as the target domain varies along with different tasks, assigning higher attention weights to the model most relevant to the current task. This adaptive prioritization of knowledge enhances context-aware reasoning, enabling more informed and effective decision-making.

Table 3: Ablation study on prototype-based world model retrieval and world-wise compound attention

(a) Prototype-based world model retrieval

| Model | SR (↑) | PS (↓) |
|---|---|---|
| WorMI-E | 48.63%±0.25% | 18.93±0.49 |
| WorMI-R | 62.04%±0.44% | 16.96±0.20 |
| WorMI | **66.12%±0.80%** | **15.17±0.08** |

(b) World-wise compound attention

| Model | SR (↑) | PS (↓) |
|---|---|---|
| WorMI-CONCAT | 47.79%±2.39% | 20.03±0.07 |
| WorMI-ADD | 56.47%±0.61% | 17.43±0.30 |
| WorMI | **66.12%±0.80%** | **15.17±0.08** |

**Ablation Study.** For the ablation study on the prototype-based world model retrieval, we compare WorMI against two variants: WorMI-E, which uses all world models for each inference, and WorMI-R, which uses a randomly se-

lected subset of world models. In both WorMI and WorMI-R, we selectively use three out of the six available world models. As shown in Table 3(a), WorMI outperforms WorMI-E and WorMI-R by 17.49% and 4.08% in SR, while reducing PS by 3.76 and 1.79 steps, respectively. These results confirm that the prototype-based retrieval effectively selects the most suitable world models at test time.

For the ablation study on the world-wise compound attention method, we compare WorMI against two variants: WorMI-CONCAT, which concatenates all world model representations, and WorMI-ADD, which instead sums these representations. As shown in Table 3(b), WorMI outperforms WorMI-CONCAT and WorMI-ADD by 18.33% and 9.65% in SR, while reducing PS by 4.86 and 1.79 steps, respectively. These improvements are attributed to the world-level cross-attention's ability to integrate relevant knowledge from the world models effectively, consistent with the domain-dependent attention outcomes shown in Figure 4.

Table 4: Impact of LLMs on WorMI and baselines

| LLM | Model | SR (↑) | PS (↓) |
|---|---|---|---|
| Llama-3.2-11B | LLM-Planner | 51.11%±0.27% | 18.33±0.08 |
| | SayCanPay | 49.28%±0.20% | 18.88±0.10 |
| | WorMI | **71.18%±0.62%** | **13.95±0.11** |
| Llama-3.2-3B | LLM-Planner | 21.45%±0.42% | 27.73±0.05 |
| | SayCanPay | 45.71%±0.59% | 19.04±0.05 |
| | WorMI | **66.12%±0.80%** | **15.17±0.08** |
| Llama-3.2-1B | LLM-Planner | 9.93%±0.27% | 27.37±0.11 |
| | SayCanPay | 42.22%±1.32% | 21.34±0.33 |
| | WorMI | **49.65%±0.83%** | **18.99±0.30** |

**Impact of LLMs.** Table 4 shows that WorMI consistently outperforms the baseline methods across LLMs of various sizes, including 1B, 3B, and 11B parameters. As the LLM size increases, WorMI capitalizes on enhanced reasoning to attain progressively larger gains, ultimately surpassing LLM-Planner by 20.07% when using the larger 11B model. In contrast, LLM-Planner's performance heavily depends on the LLM size, exhibiting substantial degradation with

Table 5: Impact of world model scale (WMs)

| Num. of WMs | SR (↑) | PS (↓) |
|---|---|---|
| 1 | 42.75%±0.50% | 20.54±0.02 |
| 2 | 62.94%±0.12% | 17.21±0.20 |
| 3 | 66.12%±0.59% | 15.17±0.08 |
| 4 | 61.96%±0.25% | 16.68±0.34 |
| 6 | 48.23%±0.25% | 18.93±0.49 |

Table 6: Performance for complex instruction scenarios

| Model | SR (↑) | GC (↑) | PS (↓) |
|---|---|---|---|
| **Long horizon instructions** | | | |
| LLM-Planner | 0.65%±0.51% | 17.16%±0.58% | 86.12±1.22 |
| SayCanPay | 6.54%±0.51% | 39.22%±0.66% | 57.87±0.28 |
| WorMI | **19.61%±0.88%** | **49.35%±0.55%** | **53.15±0.70** |
| **Multiple instructions** | | | |
| LLM-Planner | 1.31%±0.51% | 29.02%±0.08% | 85.17±0.76 |
| SayCanPay | 6.54%±0.52% | 42.32%±0.41% | 57.94±0.92 |
| WorMI | **20.26%±1.34%** | **60.62%±0.89%** | **43.49±1.23** |

the smaller 1B model. This highlights the limitation of in-context adaptation, whose benefits are maximized with larger, more capable LLMs. SayCanPay, on the other hand, maintains a relatively consistent performance across different model sizes but does not fully exploit the capabilities of larger LLMs. We speculate that its Can (affordance) and Pay (cost) models contribute more to SayCanPay's performance than the Say (reasoning) model, which relies on LLMs.

**Scalability for World Models.** Table 5 shows how varying the number of implanted world models affects WorMI's performance. While using between 2 and 4 world models yields higher SR and lower PS, employing only a single model or as many as six leads to a marked performance drop. By selectively retrieving only the most relevant world models through compound attention, WorMI can expand domain-specific knowledge without exceeding its capacity, thereby balancing scalability and performance.

**Complex Instructions.** We conduct two case studies in VirtualHome to verify the robustness of WorMI against complex task scenarios. Long-horizon instructions specify a sequence of tasks (or sub-goals) that must be completed step by step. Multiple instructions define a set of tasks to be executed concurrently, requiring integrated planning and execution for optimal completion. In both scenarios, we measure goal-conditioned success rate (GC) as the proportion of completed sub-goals alongside SR and PS, taking into account the extended, sequential nature of long-horizon instructions. As shown in Table 6, WorMI achieves both the highest SR and GC and the lowest PS for both scenarios. For long-horizon instructions, WorMI surpasses SayCanPay by 13.1% in SR, 13.13% in GC, and reduces PS by 4.72 steps, while for multiple instructions, it achieves gains of 13.72% in SR, 18.30% in GC, and reduces PS by 14.45

steps. Notably, the substantial decrease in PS for the multiple instructions highlights WorMI's effective reasoning to facilitate integrated task planning over a set of related instructions.

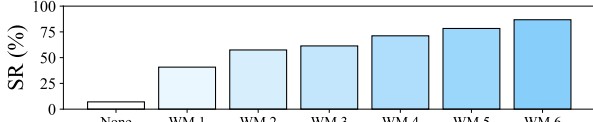

(a) Continual model implanting from WM1 to WM6

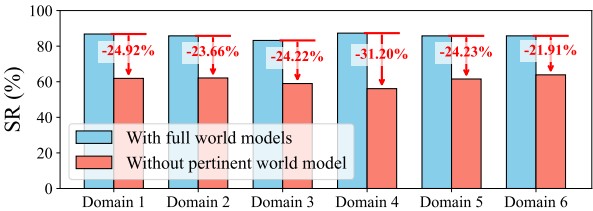

(b) Effect of removing the pertinent world model

Figure 5: Performance for continual model implanting

**Continual Model Implanting.** We consider a case of dynamic model addition and removal, which is referred to as continual model implanting. In Figure 5(a), we demonstrate how performance improves as we incrementally add new world models from WM1 to WM6. In this evaluation, we intentionally employ only the world models pertinent to the target domains, confirming that each domain-specific model provides complementary knowledge and leads to progressively better overall results. Conversely, Figure 5(b) demonstrates how removing the world models can reduce performance. We observe a machine unlearning effect by discarding specific world models no longer available, confirming the impact of lost domain-specific knowledge on overall performance. This shows WorMI's extensibility to various unlearning scenarios, where irrelevant or outdated knowledge can be seamlessly removed.

## 5. Conclusion

We presented the WorMI framework which enables an embodied policy to dynamically compose multiple world models with its fixed reasoning model at test time, thereby enhancing cross-domain adaptability. By combining the prototype-based world model retrieval with the world-wise compound attention, WorMI achieves dual-stage knowledge fusion, encompassing world-to-world knowledge integration and world-to-reasoning alignment. Evaluation results in VirtualHome and ALFWorld confirm that WorMI achieves robust adaptation to unseen domains in both zero-shot and few-shot scenarios, outperforming several LLM-based baselines.

**Limitation.** Despite its advantages, WorMI faces two key limitations. First, inferring over multiple domain-specific

world models increases computational overhead, potentially posing challenges in resource-constrained settings. Second, the framework's reasoning relies on an LLM, making its performance inherently dependent on the strengths and weaknesses of the underlying language model.

## Acknowledgement

This work was supported by Institute of Information & communications Technology Planning & Evaluation (IITP) grant funded by the Korea government (MSIT), (RS-2022-II220043 (2022-0-00043), Adaptive Personality for Intelligent Agents, RS-2022-II221045 (2022-0-01045), Self-directed multi-modal Intelligence for solving unknown, open domain problems, RS-2025-02218768, Accelerated Insight Reasoning via Continual Learning, and RS-2019-II190421, Artificial Intelligence Graduate School Program (Sungkyunkwan University)), the National Research Foundation of Korea (NRF) grant funded by the Korea government (MSIT) (No. RS-2023-00213118), IITP-ITRC (Information Technology Research Center) grant funded by the Korea government (MIST) (IITP-2025-RS-2024-00437633, 10%), IITP-ICT Creative Consilience Program grant funded by the Korea government (MSIT) (IITP-2025-RS-2020-II201821, 10%), and by Samsung Electronics.

## Impact Statement

This paper presents work whose goal is to advance the field of Machine Learning. There are many potential societal consequences of our work, none which we feel must be specifically highlighted here.

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

# A. Supplementary Proofs

## A.1. Boundedness of Prototype-based Similarity

By leveraging prototypes, the similarity between datasets can be bounded by the similarity between these prototypes. Based on the triangle inequality for the Wasserstein distance, we have

$$\delta(\mathbf{p}_i, \mathbf{p}_j) \leq \delta(\mathcal{E}_i, \mathcal{E}_j) + \delta(\mathbf{p}_i, \mathcal{E}_i) + \delta(\mathbf{p}_j, \mathcal{E}_j). \tag{A.1}$$

By optimizing (3), we obtain a minimal maximum distance $\rho$ such that

$$\forall x \in \mathcal{E}_j, \exists c \in C : \|x - c\| \leq \rho. \tag{A.2}$$

Since any point in $\mathcal{E}_i$ and $\mathcal{E}_j$ can be matched to a prototype within a distance $\rho$, the distance between the prototype sets can be bounded as

$$\delta(\mathbf{p}_i, \mathbf{p}_j) \leq \delta(\mathcal{E}_i, \mathcal{E}_j) + 2\rho. \tag{A.3}$$

# B. Environments

## B.1. VirtualHome

VirtualHome (Puig et al., 2018) is a simulation environment developed with Unity, designed to let embodied agents interact with everyday household items in order to complete given instructions. In VirtualHome, we use 20 varied house configurations each with unique room layouts and object placements, creating a complex testbed that closely mirrors real-world domestic scenarios. Within this embodied space, agents can perform a wide spectrum of actions - from picking up and moving objects to switching appliances on and off, or opening and closing doors and drawers.

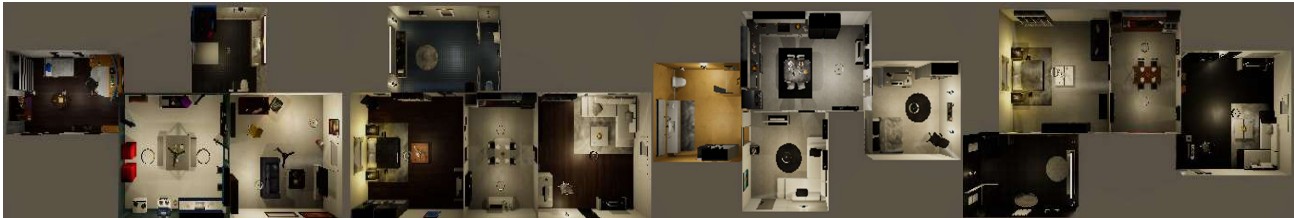

Figure A.1: Visualization of VirtualHome

(faucet, inside, bathroom), (stall, inside, bathroom), (kitchencounterdrawer, inside, kitchen), (mousemat, inside, livingroom), (bookshelf, inside, livingroom), (character, close, plum), (bedroom, adjacent, kitchen), (stove, inside, kitchen), (peach, inside, bedroom), (character, inside, bedroom), (tv, inside, livingroom), (character, hold, none), ...

Figure A.2: An example of an observation in VirtualHome

**Observations and Actions.** We use 6 action types (e.g., walk, grab, open, put, putin, and switchon) in VirtualHome, which update the graph according to the rules of VirtualHome Environments. We represent VirtualHome's graph as a list of triples, where each triple describes an object node and its relationship to another node. The details of VirtualHome actions are provided in Table A.1. Action examples are from VirtualHome Documents.

**Instructions.** There are 78 instructions consisting of 4 types of tasks. These tasks instruct the agents on what they should do in a situation. The goal of an instruction is to reach a successful state, that is represented by triples of environment graph. Details are in Table A.2.

## B.2. ALFWorld

ALFWorld is a text-driven simulation for household tasks, created to help train and evaluate agents on formulating high-level strategies from natural language instructions in an embodied context. Drawing on a series of tasks originally introduced by

Table A.1: Actions in VirtualHome

| Types | Example |
|---|---|
| Walk [Object] or Walk[room] | walk kitchen |
| Grab [Grabbable Object] | grab apple |
| Open [Object] | open fridge |
| Put [Grabbable object] [Object] | put apple table |
| PutIn [Grabbable object] [Containers] | putin apple fridge |
| SwitchOn [Switchable object] | switchon stove |

Table A.2: Instructions in VirtualHome

| Type | Amount | Example | Goal State |
|---|---|---|---|
| TurnOn | 9 | Turn on tv
Turn on radio
Turn on microwave | [(tv, is, on)]
[(radio, is, on)]
[(microwave, is, on)] |
| Open | 7 | Open cabinet
Open dishwasher
Open microwave | [(cabinet, is open)]
[(dishwasher, is, open)]
[(microwave, is, open)] |
| PutOn | 30 | Put apple on desk
Put clock on sofa
Put bananas on microwave | [(apple, on, desk)]
[(clock, on, sofa)]
[(bananas, on, sofa)] |
| PlaceIn | 32 | Place towel in closet
Place paper in bookshelf
Place plum in fridge | [(towel, inside, closet)]
[(paper, inside, bookshelf)]
[(plum, inside, fridge)] |

the ALFRED benchmark, it maintains long task sequences and unidirectional state changes—key elements that give the environment a high degree of authenticity and practicality.

**Observations and Actions.** ALFWorld is a combination of text and visual simulation. When a scenario begins, the agent's current observation and the assigned task are displayed to the user in text form. The goal of the game is to reach a certain successful state by entering the appropriate text commands. A description of the game's observation–action sequence is provided in Figure A.3, and the list of possible actions can be found in Table A.3.

-= Welcome to TextWorld, ALFRED! =-
Observation: You are in the middle of a room. Looking quickly around you, you see a armchair 1, a coffeetable 1, a garbagecan 1, a shelf 14, a shelf 13, a shelf 12, a shelf 11, a shelf 10, a shelf 9, a shelf 8, a shelf 7, a shelf 6, a shelf 5, a shelf 4, a shelf 3, a shelf 2, a shelf 1, a sofa 1, a tvstand 2, and a tvstand 1.
Your task is to: put a pillow in armchair.
Action: look
Observation: You are in the middle of a room. Looking quickly around you, you see nothing.
Action: go to sofa 1
Observation: You arrive at sofa 1. On the sofa 1, you see a box 2, a creditcard 3, a pillow 2, a pillow 1, and a remotecontrol 2.
...

Figure A.3: An example of a trajectory in ALFWorld

**Instructions.** There are 6 main instruction types in ALFWorld, which imply the overall content that an agent must perform in a scenario. By combining numerous objects, receptacles, and rooms with one instruction, 3,554 scenarios are created. The details are explained in Table A.4.

Table A.3: Actions in ALFWorld.

| Type | Example |
|------|---------|
| Goto [Receptacle Object] | Goto dresser |
| Open [Receptacle Object] | Open microwave |
| Close [Receptacle Object] | Close microwave |
| Pickup [Object] [Receptacle Object] | Take alarmclock from dresser |
| Put [Object] [Receptacle Object] | Put book in/on dresser |
| Heat [Object] [Receptacle Object] | Heat bread with microwave |
| Cool [Object] [Receptacle Object] | Cool cup with fridge |
| Clean [Object] [Receptacle Object] | Clean kettle with sinkbassin |
| Slice [Object] [Instrument Object] | Slice bread with knife |
| Examine [Object] | Examine clock |
| Examine [Receptacle Object] | Examine countertop |

Table A.4: Instructions in ALFWorld

| Type | Example |
|------|---------|
| Pick & Place | Place a box with keys in it on the coffee table. |
| Pick Two & Place | Place two wine bottles in a bin. |
| Clean & Place | Put a clean bar of soap in the drawer. |
| Heat & Place | Put a warmed slice of tomato on the table. |
| Cool & Place | Place a chilled potato slice inside the microwave. |
| Examine & in Light | Inspect a racket with a lamp on a desk. |

# C. Implementation Details

## C.1. Baselines

**ZSP** (Huang et al., 2022a) is a zero-shot policy approach that directly applies a pre-trained model to new domains without any adaptation procedure. This baseline serves as a reference to assess the improvements achieved through our WorMI. It uses a single LLM (Llama-3.2-3B-Instruct), and takes an observation from the environment, then injects it into the prompt described in Figure A.4 and A.5. So. it takes the prompt and generates an action step by step. For implementation, we refer to the opensource [1].

**LLM+FT** is a representative baseline that demonstrates adaptation through fine-tuning on limited domain-specific data. By comparing against this approach, we evaluate the efficiency and effectiveness of WorMI relative to fine-tuning in cross-domain adaptation. In the few-shot adaptation scenario, we additionally train the fine-tuned LLM with the few-shot data sampled from the target domain.

**LLM-Planner** (Song et al., 2023) is an embodied planner that leverages LLMs for reasoning and task execution within the environment. It employs in-context learning to generate high-level plans and adapt them based on current observations. This baseline allows us to evaluate the gains achieved by WorMI over the in-context learning method. For evaluation, we employ the DPR-based sentence embedding model for the retriever, which uses the same dataset employed in the training of other approaches requiring additional fine-tuning (e.g., LLM+FT, SayCanPay, WorMI). Furthermore, for the few-shot scenarios, we add few-shot data examples into the retriever. For implementation, we refer to the opensource [2].

**SayCanPay** (Hazra et al., 2024) is a state-of-the-art reinforcement learning-based planning approach that incorporates pre-trained skills to assess feasibility and employs heuristic cost minimization. By comparing against SayCanPay, we aim to demonstrate the planning efficiency of WorMI. It uses three models under the same prompts: (1) We use the Llama-3.2-3B model as a Say model, (2) We use optimal affordance from the environment for a Can model, and (3) We train the Pay model based on the Llama-3.2-1B. For implementation, we refer to the opensource [3]. The hyperparameter settings of SayCayPay are in Table A.5.

---

[1]https://github.com/huangwl18/language-planner
[2]https://github.com/OSU-NLP-Group/LLM-Planner
[3]https://github.com/RishiHazra/saycanpay

The hyperparameter settings of baselines are summarized in Table A.5. The prompts we use are described in Figure A.4 and A.5.

Table A.5: Hyperparameter settings and configurations of baselines

| Hyperparameter | Value |
| --- | --- |
| Trainable model (LLM+FT, and Pay model in SayCayPay) | Llama-3.2-1B |
| Reasoning model (ZSP, LLM-Planner, and Say model in SayCayPay) | Llama-3.2-3B |
| Batch size | 4 |
| Gradient steps | 200 |
| Learning rate scheduler | cosine |
| Initial learning rate | $5 \times 10^{-5}$ |
| Learning rate (for few-shot learning) | $1 \times 10^{-6}$ |
| Temperature (both of Llama-3.2-1B and Llama-3.2-3B) | 1.0 |

---

**[System]**
You are a home robot agent. You can use 6 skills, (*walk [object or room], grab [object], switch [object], open [object], putin [target object], put [target object]*). You should return only a skill after "*Action*:". Room: *livingroom, bathroom, kitchen, bedroom.*

**[User]**
Instruction: {*instruction*}
Observation: {*observation*}
Action:

---

Figure A.4: System prompt in VirtualHome

---

**[System]**
You are a home robot agent. You can use 10 skills, (*go to [object], take [object] from [object], put [object] on [object], open [object], close [object], toggle [object], heat [object] with [object], cool [object] with [object], clean [object] with [object], look*). You should return only a skill after "*Action*:". Room: *livingroom, bathroom, kitchen, bedroom.*

**[User]**
Instruction: {*instruction*}
Observation: {*observation$_0$*}
Action: {*action$_0$*}
Observation: {*observation$_1$*}
Action: {*action$_1$*}
...
Action:

---

Figure A.5: System prompt in ALFWorld

## C.2. WorMI (Ours)

Our WorMI framework integrates two key methods into an adaptive, composable policy structure tailored for LLM-based embodied agents, facilitating the test-time, dual-stage composition of world models. (a) A **prototype-based world model retrieval** method selectively activates only a set of relevant world models. To determine relevance, each model's similarity to the current target domain is measured using object-wise state embeddings and clustering outcomes derived from trajectory-based prototypes. This ensures a more robust and interpretable adaptation process across diverse domains, particularly in

zero-shot or few-shot settings. (b) A **world-wise compound attention** method effectively integrates the world models with the reasoning model by selectively combining the most pertinent knowledge from the set. This facilitates effective and efficient policy adaptation during test-time execution. The interplay of these two methods enables the agent to dynamically compose and contextualize domain-specific knowledge in its policy across domains, through coherent integration and alignment of world models and a reasoning model. The hyperparameter settings of WorMI are in Table A.6. Algorithm 1 shows the entire procedure of the world model implanting framework.

---

**Algorithm 1** WorMI Framework

---

1: World models $\{M_1, \ldots, M_N\}$, Reasoning model $\pi_R$, Datasets $\mathcal{D}_1, ..., \mathcal{D}_N$, learning rate $\alpha, \beta$
2: World-wise compound attention $C_\theta$ for policy $\pi_\theta$
3: // Training world-wise compound attrition via meta-learning
4: Initialize parameters $\theta$ randomly
5: Sample the subset of world models $\mathbf{M}_1, \cdots \subset \{M_1, ..., M_N\}$ and associated datasets $\mathbf{D}_1, \cdots \subset \{\mathcal{D}_1, ..., \mathcal{D}_N\}$
6: **for** meta update steps $= 1, 2, ..., \lambda_M$ **do**
7:     **for** each $\mathbf{M}_j$ and $\mathbf{D}_j$ **do**
8:         $\theta_j \leftarrow \theta$
9:         Implant the world model $\pi_{\theta_j} = C_{\theta_j}(\mathbf{M}_j, \pi_R)$
10:         **for** iteration$=1, 2, ..., \lambda_I$ **do**
11:             Sample batch $\mathcal{B}$ from Dataset $\mathbf{D}_j$
12:             $\theta_j \leftarrow \theta_j - \alpha \nabla_{\theta_j} \mathcal{L}(\theta_j, \mathcal{B})$ in (9)
13:         **end for**
14:     **end for**
15:     $\theta \leftarrow \theta + \beta \cdot \frac{1}{M} \sum_{j=1}^M (\theta_j - \theta)$
16: **end for**
17:
18: // Test-time adaptation with the prototype-based world model retrieval and world-wise compound attention
19: Object detection model $\Phi_D$, Embedding model $\Phi_E$, Environment $env$
20: **for** $j = 1, 2, ..., N$ **do**
21:     $\mathcal{E}_j = \{\Phi_E(o) \,|\, \{o_1, ..., o_N\} = \Phi_D(s), s \in \mathcal{D}_j\}$
22:     Optimize prototype $\mathbf{p}_j$ by (3)
23: **end for**
24: $t = 0, s_t = env.\text{reset}()$
25: **loop**
26:     $\mathcal{E}_j = \{\Phi_E(o) \,|\, \{o_1, ..., o_N\} = \Phi_D(s), s_t\}$
27:     Optimize prototype $\mathbf{p}$ by (3)
28:     $\mathbf{M}_{\text{ret}} = \{M_j \,\big|\, j \in \text{TopK}(\{-\delta(\mathbf{p}_j, \mathbf{p})\}_{j=1}^N, \ K)\}$
29:     $\pi_\theta = C_\theta(\mathbf{M}_{\text{ret}}, \pi_R), \ a_t = \pi_\theta(s_t)$
30:     $s_{t+1} = env.\text{step}(a_t), t \leftarrow t + 1$
31: **end loop**

---

Table A.6: Hyperparameter settings and configurations of WorMI

| Parameter | Value |
|---|---|
| **Learning world models** $M_1, ..., M_N$ | |
| Base model | Llama-3.2-1B |
| Batch Size | 4 |
| Gradient steps | 2000 |
| Learning Rate Scheduler | cosine |
| Learning Rate | $3 \times 10^{-5}$ |
| Temperature | 1.0 |
| Intermediate connection layer | [13, 27] |
| **Learning compound attention** | |
| Reasoning model $\pi_R$ | Llama-3.2-3B |
| Batch Size | 4 |
| Meta update steps $\lambda_M$ | 8 |
| Inner-loop gradient steps $\lambda_I$ | 30 |
| Learning Rate Scheduler | cosine |
| Learning Rate $\alpha$ | $1 \times 10^{-5}$ |
| Meta learning Rate $\beta$ | $1 \times 10^{-1}$ |
| Temperature | 1.0 |
| Learning Rate (for few-shot learning) | $1 \times 10^{-5}$ |
| Intermediate connection layer | [13, 27] for reasoning model [7,15] for world models |
| **Prototype-based world model retrieval** | |
| The number of embeddings in prototype k | 15 |
| The number of world models N | 6 |
| The number of retrieved world models K | 3 |

# D. Additional Analysis

## D.1. Analysis of world model usage

We categorize different cases to analyze the world-level attention map.

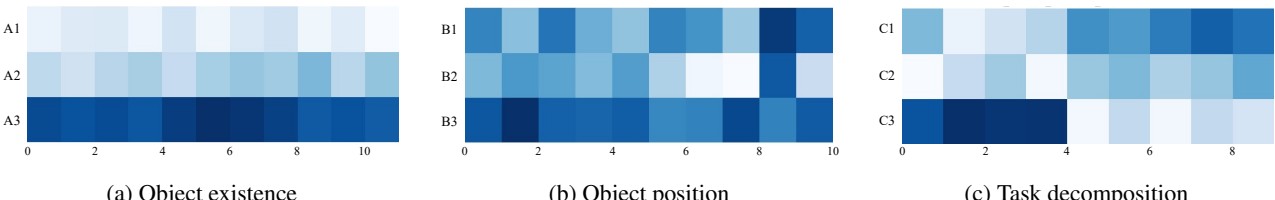

(a) Object existence  (b) Object position  (c) Task decomposition

Figure A.6: Analysis for the world-level attention value in various cases

**Case 1. Object existence.** First, we examine how the attention changes when a particular object exists in only one world model. For example, Figure A.6(a) illustrates the scenario where the object "pie" appears only in domain A3, and not in domains A1 or A2, under the instruction "Place pie in microwave." The attention value remains high in world models from domain A3 throughout all steps of the episodes, demonstrating that the world-wise cross-attention effectively focuses on the world model containing the relevant information.

**Case 2. Object position.** We examined how the world model operates when the positions of objects differ. For instance, in the task "Put mug on coffeetable," the target domain places the mug in the livingroom and the coffeetable in the bedroom. In domains B1 and B2, the mug is located in the kitchen while the coffeetable remains in the bedroom. In contrast, in domain B3, both the mug and the coffeetable are situated in the livingroom. As shown in Figure A.6(b), initially, high attention values are assigned to B3. Over time, attention values for world models from domains B1 and B2 increase. This behavior demonstrates that world-level cross-attention selectively utilizes the world models containing the relevant information needed to successfully execute the task.

**Case 3. Task decomposition.** We analyzed scenarios where tasks are partially present in each world model. For example,

when performing the task "Place plum in cabinet" in the target domain, the world model from domain C1 is trained with data related to cabinet tasks (e.g., "Place mug in cabinet"), while the world model from domain C3 is trained with data related to plum tasks (e.g., "Place plum in fridge"). The word model from domain C2 is less relevant. As shown in Figure A.6, during the process of picking up the plum, the world model from domain C3 exhibits a high attention value. When placing the plum in the cabinet, the world model from domain C1 shows a high attention value. This demonstrates that through knowledge integration and alignment in the world-wise compound attention, WorMI sequentially utilizes partially learned task information, achieving high performance even on unseen tasks.

### D.2. Multi-modal agents

We consider multi-modality is critical for real robotic systems, which often rely on various sensor inputs. Our WorMI framework is designed to support this by allowing the reasoning model and individual world models to come from different modalities.

The table below shows the performance of multi-modal WorMI, which employs a VLM as its reasoning model, using both text and image states in VirtualHome. Multi-modal WorMI exhibits only a slight performance drop compared to WorMI, demonstrating the applicability for multi-modal experiment setups. Additionally, there is certainly room for improvement of Multi-modal WorMI, as we do not have enough time to optimize the hyperparameters. Even so, our approach still demonstrates superior performance compared to the baselines. We will include these experimental results in the final version.

Table A.7: Performance comparisons for Multi-modal WorMI and WorMI

| Model | SR ($\uparrow$) | PS ($\downarrow$) |
|---|---|---|
| Multi-modal WorMI | 57.65% | 17.21 |
| WorMI | 66.12% | 15.17 |

### D.3. Analysis of resource usage

Below table shows the inference times and memory usage among the baselines and our WorMI. We use LLaMA-3.2-11B as the reasoning model and LLaMA-3.2-1B for the world models. The same model configurations are also used for all baselines. LLM-Planner requires a longer prompt for in-context examples, leading to increased inference time. SayCanPay utilizes three separate models and repetitive infers for the action log probabilities, which need further inference time. WorMI uses relatively smaller world models and selects only the K most relevant ones rather than using all of them. Additionally, WorMI keeps each domain-specific world model much smaller than the main reasoning model, making it easier to scale up the number of world models.

Table A.8: Inference time and memory for comparisons and WorMI

| | Inference Time | Memory |
|---|---|---|
| LLM+FT, ZSP | 298 ms | 21877 MiB |
| LLM-Planner | 401 ms | 21877 MiB |
| SayCanPay | 609 ms | 46230 MiB |
| WorMI (K=2, N=4) | 339 ms | 30020 MiB |
| WorMI (K=2, N=6) | 348 ms | 33445 MiB |
| WorMI (K=3, N=6) | 385 ms | 33445 MiB |

### D.4. Robustness and scalable analysis for prototype-based retrieval

In Table 3(a), we compare a random selection strategy (WorMI-R) to our prototype-based retrieval approach, illustrating that incorrect retrieval slightly degrades performance. To investigate this more thoroughly, we conducted two additional experiments. First, in Table A.9 we increased the proportion of adversarial world models by replacing some with untrained Llama-3.2-1B models. With prototypes unchanged, the retrieval could not distinguish adversarial from valid models. At lower proportions, performance stayed relatively stable, but it declined sharply once the adversarial ratio exceeded a certain threshold. This indicates that our compound attention helps filter out adversarial models, maintaining robustness unless a critical mass of them is adversarial.

Table A.9: Robustness of prototype-based retrieval under adversarial world model settings

| Adv. ratio | SR (↑) | PS (↓) |
|---|---|---|
| 0% | 66.12% | 15.17 |
| 16% | 66.67% | 14.96 |
| 33% | 58.58% | 16.00 |
| 50% | 38.03% | 21.42 |

We also scaled the number of world models from N=6 to N=12. As shown in Table A.10, WorMI-R suffers more from incorrect retrieval as N grows, whereas our prototype-based retrieval remains relatively resilient.

Table A.10: Scalability of prototype-based retrieval across varying world model pool sizes.

| Model | SR (↑) | PS (↓) |
|---|---|---|
| WorMI-R (N=6) | 62.04% | 16.96 |
| WorMI (N=6) | **66.12%** | **15.17** |
| WorMI-R (N=12) | 51.17% | 19.22 |
| WorMI (N=12) | **66.51%** | **14.90** |

Overall, these results confirm that suboptimal retrieval or misleading world models can lower performance, but WorMI's world-to-world knowledge integration in compound attention provides robust outcomes. Although adversarial models do pose a risk, our retrieval method and compound attention mitigate their impact, unless the majority of models are adversarial.

### D.5. Analysis of prototypes in world model retrieval

The Table A.11 shows the results comparing prototype retrieval and full retrieval (WorMI-P). The performance difference between the two is minimal, yet prototype retrieval significantly reduces inference time.

Table A.11: Effect of prototype on world model retrieval

| Model | SR (↑) | PS (↓) | Inference Time |
|---|---|---|---|
| WorMI-P | 66.54% | 15.04 | 811ms |
| WorMI | 66.12% | 15.17 | 385ms |

### D.6. Analysis of multiple world models

The Table A.12 shows that a performance comparison between WorMI and the variant that uses only the single most relevant world model without fusion (WorMI-F). For seen tasks and scenes, a single world model suffices as it captures the domain knowledge. In contrast, for unseen tasks and scenes, combining multiple world models via world-to-world integration is advantageous since no single model fully represents the unseen domain.

Table A.12: Effect of multiple world models in model implanting

| Model | SR (↑) | PS (↓) |
|---|---|---|
| Unseen tasks & scenes | | |
| WorMI-F | 42.75% | 20.54 |
| WorMI | 66.12% | 15.17 |
| Seen tasks & scenes | | |
| WorMI-F | 82.72% | 11.16 |
| WorMI | 85.78% | 10.76 |

