# OpenReview forum: "World Model Implanting for Test-time Adaptation of Embodied Agents"
_ICML.cc/2025/Conference — ICML 2025 poster_

### Official Review · Reviewer_sgRi · 2025-03-10

**Overall Recommendation:** 3

**Summary:**

This paper proposes a world model implanting framework to augment the LLM-based agents. The world models are learned using domain/task-specific data to capture the in-domain characteristics, serving as domain experts. With these world models, this paper introduces a prototype-based retrieval method together with an attention-based knowledge integration to allow test-time adaptation. This framework also applies a meta-learning objective to improve the adaptability to unseen tasks/domains. Experiments are conducted on Virtual-Home and ALFWorld benchmarks.

**Claims And Evidence:**

This paper claims that the proposed WorMI framework is able to generalize to unseen domains in zero-shot or few-shot manners by leveraging the most relevant information from the seen world models. However, how does this method guarantee that the knowledge of the seen world models will benefit the target task? If the target domain/task is highly out-of-the-training distributions, how does this paper handle this case?

**Essential References Not Discussed:**

To the best of my knowledge, the references are sufficiently covered.

**Experimental Designs Or Analyses:**

I have checked the Experiments section. For the meta-learning, this paper does not provide an ablation study or analysis to assess the effectiveness of this objective. The improvement of generalizability contributed by meta-learning remains unclear.

**Methods And Evaluation Criteria:**

The proposed method applies off-the-shelf models to extract abstract state features. However, it might be risky to discard crucial visual/structural information about the environment. In addition, without being grounded in the visual environment, how does this paper ensure the LLM provides accurate actions without potential hallucination issues?

**Other Comments Or Suggestions:**

Please refer to the issues raised above.

**Other Strengths And Weaknesses:**

Overall, this paper is well-written and easy to follow. For the weaknesses, please refer to the issues raised above.

**Questions For Authors:**

Please refer to the issues raised above.

**Relation To Broader Scientific Literature:**

The main contribution compared with previous LLM-based embodied agents is the integration of the domain-specific models, which provide domain knowledge to augment the LLM.

**Theoretical Claims:**

No theoretical claims are provided in the main paper.

---

> ### Author Rebuttal · Authors · 2025-04-01
>
> We sincerely appreciate your detailed comments. We will include the following experimental results and clarifications in the final version.
>
> > Q1. How does this method guarantee that the knowledge of the seen world models will benefit the target task? If the target domain/task is highly out-of-the-training distributions, how does this paper handle this case?
>
> Our method guarantees that the knowledge of seen world models benefits the target task by combining two key components.
> First, the prototype-based retrieval selects only those world models that were trained on data with distributions similar to the target environment, ensuring that the most relevant domain-specific information is leveraged.
> Second, our compound attention model integrates the selected world models with the reasoning model on two levels, world-level integration and reasoning-level alignment, while meta-learning is used to adapt effectively to diverse world model combinations. This process not only creates a robust, domain-specific composite model but also aligns the integrated knowledge with the reasoning model to enhance decision-making.
> This approach is validated by our results in Table 1, where our method outperforms baselines even in unseen domains, confirming that the aligned knowledge is indeed beneficial for the target task.
>
> Moreover, if the environment is completely unrelated to any available world model, the agent can still utilize the reasoning model’s common knowledge via world-to-reasoning alignment, maintaining at least a baseline level of performance. Please refer to our response to Reviewer SQ6Q, Q4 for details on the adversarial world models experiments, which involve models that are highly out-of-training distribution.
> Of course, in highly out-of-distribution scenarios that lie well beyond our assumptions, the success rate may drop significantly.
>
> > Q2. It might be risky to discard crucial visual/structural information about the environment. In addition, without being grounded in the visual environment, how does this paper ensure the LLM provides accurate actions without potential hallucination issues?
>
> In VirtualHome and ALFWorld, the environment provides text-based observations that include visual and structural information for performing tasks.
> These text-based outputs have been widely used in prior work [1, 2, 3, 4, 5] and the agent can select suitable actions without discarding crucial spatial or visual context.
> Moreover, our WorMI framework is designed to be modular with respect to input modalities: the reasoning model and individual world models can each handle different data forms.
>
> The table below shows the performance of multi-modal WorMI, which employs a VLM as its reasoning model, using both text and image states in VirtualHome.
> Multi-modal WorMI exhibits only a slight performance drop compared to WorMI, demonstrating the applicability for multi-modal experiment setups.
> Additionally, there is certainly room for improvement of Multi-modal WorMI, as we do not have enough time to optimize the hyperparameters.
> Even so, our approach still demonstrates superior performance compared to the baselines.
> We will include these experimental results in the final version.
>
> | Model             | SR (↑) | PS (↓) |
> |------------------|--------|--------|
> | Multi-modal WorMI| 57.65% | 17.21  |
> | WorMI            | 66.12% | 15.17  |
>
> [1] Huang, Wenlong, et al. "Language models as zero-shot planners: Extracting actionable knowledge for embodied agents." ICML 2022.
>
> [2] Song, Chan Hee, et al. "Llm-planner: Few-shot grounded planning for embodied agents with large language models." ICCV 2023.
>
> [3] Hazra, Rishi, Pedro Zuidberg Dos Martires, and Luc De Raedt. "Saycanpay: Heuristic planning with large language models using learnable domain knowledge." AAAI 2024.
>
> [4] Singh, Ishika, et al. "Progprompt: Generating situated robot task plans using large language models.", ICRA 2023.
>
> [5] Yoo, Minjong, et al. "Exploratory retrieval-augmented planning for continual embodied instruction following.", NeurIPS 2024.
>
> > Q3.  For the meta-learning, this paper does not provide an ablation study or analysis to assess the effectiveness of this objective. The improvement of generalizability contributed by meta-learning remains unclear.
>
> The table below shows an ablation study on meta-learning for unseen tasks and scenes. WorMI-M is a variant that learns world model composition sequentially instead of using meta-learning.
> Compared to the WorMI, WorMI-M exhibits lower performance. This indicates that meta-learning equips our framework to better handle world model combinations it may not have encountered during training, thereby improving generalizability.
> We will include these results in the final version.
>
> | Model    | SR (↑) | PS (↓) |
> |----------|--------|--------|
> | WorMI-M  | 53.31% | 18.23  |
> | WorMI    | 66.12% | 15.17  |

---

> > ### Comment · Reviewer_sgRi · 2025-04-02
> >
> > Thanks to the authors for providing the rebuttal. I've read the author's response and comments from other reviewers. I have no further questions at this time. I will increase my original rating to 3.

---

> > > ### Author Response · Authors · 2025-04-02
> > >
> > > We deeply appreciate the reviewer’s insightful and constructive feedback. We are encouraged by the comments noting that our paper is well-written and easy to follow. The reviewer's insights on the experiments with highly out-of-distribution data, multi-modal evaluation, and the ablation study on meta-learning are extremely valuable. We will include these suggestions in the final version. Thank you once again!

---

### Official Review · Reviewer_SQ6Q · 2025-03-11

**Overall Recommendation:** 3

**Summary:**

The paper introduces WorMI, a framework designed to improve the adaptability of embodied AI agents across diverse and unseen domains at test time, without requiring extensive retraining or additional data collection. Experiments on the VirtualHome and ALFWorld benchmarks demonstrate that WorMI outperforms state-of-the-art LLM-based embodied agents in zero-shot and few-shot adaptation. The framework also supports continual model implanting, allowing new world models to be added or removed flexibly.

**Claims And Evidence:**

WorMI’s effectiveness in zero-shot and few-shot adaptation is strongly supported. In zero-shot tests, WorMI outperforms all baselines on unseen tasks and scenes.

The paper also claims advantages of the prototype-based world model retrieval mechanism, which selects relevant domain-specific models at test time. This claim is backed by an ablation study: using WorMI’s prototype retrieval to pick a few pertinent world models yields better results than naive alternatives.

The authors further claim that WorMI’s design generalizes across unseen domains. This is reflected in the experimental setup: in VirtualHome and ALFWorld, “unseen” test scenarios involve new tasks and/or environments not encountered in training.

**Essential References Not Discussed:**

Some test-time adaptation papers should be added to the related work section for a more comprehensive discussion:

Test-time adaptation: Tent: Fully Test-Time Adaptation by Entropy Minimization
Test-Time Classifier Adjustment Module for Model-Agnostic Domain Generalization
Test-Time Training with Self-Supervision for Generalization under Distribution Shifts

**Experimental Designs Or Analyses:**

The experimental setup appears mostly sound and well-motivated for the problem of test-time adaptation in embodied AI. The study evaluates WorMI on VirtualHome (a 3D household task simulation) and ALFWorld (a text-based embodied environment). These benchmarks are reasonable choices because they provide structured tasks, unseen environments, and diverse domain shifts. However, both are simulations, meaning results may not fully translate to real-world robotics applications.

The paper compares WorMI against four baselines, covering a range of adaptation approaches. These baselines are appropriate and diverse, making the comparisons meaningful.

The experiments are well-described and appear repeatable, as dataset splits, baselines, and architectures are clearly outlined.

**Methods And Evaluation Criteria:**

The WorMI framework logically addresses the problem of test-time adaptation for embodied agents. It does so by allowing the agent to dynamically retrieve and integrate domain-specific knowledge at inference time, rather than requiring retraining.

However, there are several limitations of the proposed method.

It introduces computational overhead due to retrieving and integrating multiple world models at test time, which may not scale well in resource-constrained settings.

The framework heavily depends on large language models, making its performance sensitive to the LLM’s limitations, such as biases and hallucinations.

Adaptation to dynamic environments is limited, as the approach primarily focuses on structured, predefined world models rather than handling real-time environmental changes.

The lack of explicit robustness tests makes it unclear how well WorMI handles incorrect retrieval or misleading world models.

**Other Comments Or Suggestions:**

The paper is well-written, with clear explanations and a logical structure. The writing is concise and easy to follow, making the technical contributions accessible.

The figures are well-designed and effectively illustrate key concepts, particularly the retrieval mechanism and world-wise compound attention.

**Other Strengths And Weaknesses:**

Strengths:

The paper presents an innovative approach to test-time adaptation by dynamically retrieving and fusing world models to enhance embodied agent reasoning.

The empirical results demonstrate strong improvements over baselines in both zero-shot and few-shot adaptation scenarios.

Weaknesses:

Computational or memory efficiency is not analyzed, even though WorMI requires additional computation for retrieval and fusion.

The study does not provide qualitative insights into failure cases, making it unclear why adaptation fails in some scenarios.

The paper claims that WorMI is scalable, but it only evaluates retrieval from a small number of world models. It does not test how well the approach scales when hundreds of world models must be retrieved dynamically.

There is no explicit analysis of what happens when the retrieval mechanism selects irrelevant world models, which could be a key failure mode in complex environments.

The ablations confirm that both prototype retrieval and compound attention contribute positively, but they do not explore simpler alternatives like selecting the most relevant world model without fusion, which could reveal whether the full attention mechanism is necessary.

**Questions For Authors:**

How well does WorMI scale when retrieving from a large number of world models (e.g., 50–100 instead of ≤6)? Have you tested how retrieval accuracy or computational cost changes as the number of world models increases?

What happens when the retrieval mechanism selects an irrelevant or suboptimal world model? Is there a mechanism for detecting and correcting retrieval errors at test time?

Have you considered evaluating WorMI in a real-world robotic setting instead of simulation? What are the main challenges in transferring the approach from VirtualHome and ALFWorld to physical environments?

**Relation To Broader Scientific Literature:**

The concept of world models has been widely explored in robotics and reinforcement learning. Prior work, such as World Models by Ha & Schmidhuber (2018), introduced learned world models to enable agents to simulate future states and make more efficient decisions.
The WorMI framework extends world models by allowing multiple world models to be retrieved and fused dynamically at test time. Unlike prior methods where a single learned world model guides decision-making, WorMI selects the most relevant world models per task, combining them with an LLM for reasoning.

Test-time adaptation aims to help models generalize to new environments without retraining. WorMI introduces a retrieval-based approach to test-time adaptation. Instead of updating a model’s parameters during test time, it retrieves and fuses world models dynamically, reducing the need for computationally expensive online updates.

WorMI follows a modular learning paradigm by treating each world model as a domain-specific knowledge module that can be combined dynamically. (Modular Multitask Reinforcement Learning with Policy Sketches)

**Theoretical Claims:**

The paper includes a key theoretical component related to prototype-based world model retrieval. It provides a proof that the distance between prototype sets can serve as a bounded proxy for the distance between full datasets. While the proof shows a bound, it does not quantify how much information is lost by using prototypes instead of full embeddings. The paper does not provide experimental results comparing prototype retrieval vs. full retrieval to validate this bound in practice.

---

> ### Author Rebuttal · Authors · 2025-03-31
>
> We sincerely appreciate your detailed comments. We will include references, the following experimental results, and additional clarifications in the final version.
> > Q1. Experimental results for inference time and memory usage
>
> Below shows the inference times and memory usages among the baselines and WorMI.
> A detailed explanation is provided in response to Reviewer zfbw, Q3.
> | |Inference Time|Memory|
> |-|-|-|
> |LLM+FT, ZSP|298ms|21877MiB|
> |LLM-Planner|401ms|21877MiB|
> |SayCanPay|609ms|46230MiB|
> |WorMI(K=2,N=4)|339ms|30020MiB|
> |WorMI(K=2,N=6)|348ms|33445MiB|
> |WorMI(K=3,N=6)|385ms|33445MiB|
> > Q2. Clarification on the limitations due to utilizing LLMs
>
> WorMI leverages domain-specific world models to reduce hallucination and bias instead of relying solely on the LLM’s own knowledge. As shown in Table 4, it remains robust even with a smaller LLM while keeping expansion costs low. However, completely eliminating LLM weaknesses remains challenging. These limitations are explicitly addressed in the main text.
> > Q3. Experimental results for real-time environmental change
>
> WorMI composes pre-trained models at test time, forming a composite world model that adapts to unseen or dynamic environments.
> Below are the results under real-time environment changes, such as shifting object locations or state changes over time.
> |Model|SR (↑)|PS (↓)|
> |-|-|-|
> |LLM-Planner|45.35%|20.16|
> |SayCanPay|41.33%|21.91|
> |WorMI|58.36%|19.33|
> > Q4. Robustness and scalable tests for prototype-based retrieval
>
> In Table 3(a), we compare a random selection strategy (WorMI-R) to our prototype-based retrieval approach, illustrating that incorrect retrieval slightly degrades performance. To investigate this more thoroughly, we conducted two additional experiments.
>
> First, we increased the proportion of adversarial world models by replacing some with untrained Llama-3.2-1B models. With prototypes unchanged, the retrieval could not distinguish adversarial from valid models. At moderate proportions of adversarial world models, performance remains stable, but it drops sharply when their proportion becomes too high.
> |Adv. ratio|SR(↑)|PS(↓)|
> |-|-|-|
> |0%|66.12%|15.17|
> |16%|66.67%|14.96|
> |33%|58.58%|16.00|
> |50%|38.03%|21.42|
>
> We also scaled the number of world models from N=6 to N=12. As shown in the table below, WorMI-R suffers more from incorrect retrieval as N grows, whereas our prototype-based retrieval remains relatively resilient.
> |Model|SR(↑)|PS(↓)|
> |-|-|-|
> |WorMI-R(N=6)|62.04%|16.96|
> |WorMI(N=6)|66.12%|15.17|
> |WorMI-R(N=12)|51.17%|19.22|
> |WorMI(N=12) |66.51%|14.90|
>
> Overall, these results show that while suboptimal retrieval or misleading world models can reduce performance, WorMI’s compound attention integrating world-to-world knowledge remains robust, unless most models are adversarial. We will include these results, along with an experiment using more world models, in the final version.
> > Q5. Experimental results comparing prototype retrieval vs. full retrieval
>
> The table below shows the results comparing prototype retrieval and full retrieval (WorMI-P). There is almost no performance difference, but prototype retrieval significantly reduces inference time.
> |Model|SR(↑)|PS(↓)|Inference Time|
> |-|-|-|-|
> |WorMI-P| 66.54%|15.04|811ms|
> |WorMI|66.12%|15.17|385ms|
> > Q6. Qualitative analysis
>
> One notable failure case arises when there is no relevant world model for the unseen domain.
> For instance, if the instruction is “Place the slipper in the closet", but the agent has never encountered a slipper in any seen domain, the success rate drops and the agent might attempt actions for a similar object (e.g., “towel”).
> It is because world-to-reasoning alignment training encourages reliance on the knowledge derived from the world model, the agent struggles in scenarios where it must rely solely on the reasoning model’s common knowledge.
> In future work, we plan to explore more flexible architectures and training methods for controlling the degree of utilizing world models.
> > Q7. Comparison with using most relevant world models
>
> The table below shows that a performance comparison between WorMI and the variant that uses only the single most relevant world model without fusion (WorMI-F).
> For seen tasks and scenes, a single world model suffices as it captures the domain knowledge. In contrast, for unseen tasks and scenes, combining multiple world models via world-to-world integration is advantageous since no single model fully represents the unseen domain.
> |Model|SR(↑)|PS(↓)|
> |-|-|-|
> |Unseen tasks & scenes|
> |WorMI-F|42.75%|20.54|
> |WorMI|66.12%|15.17|
> |Seen tasks & scenes|
> |WorMI-F|82.72%|11.16|
> |WorMI|85.78%|10.76|
> > Q8. Challenges for real world settings
>
> We consider multi-modality is critical for real robotic systems, which often rely on various sensor inputs. Our WorMI framework is designed to support this by allowing the reasoning model and individual world models to come from different modalities. We add a multi-modal experiment in response to Reviewer MES3, Q3.

---

> > ### Comment · Reviewer_SQ6Q · 2025-04-03
> >
> > The authors have addressed most of my questions. I will keep my current rating for this paper.

---

> > > ### Author Response · Authors · 2025-04-03
> > >
> > > We are truly grateful for the reviewer's insightful and constructive feedback. We are encouraged by the comments noting that our approach is innovative, demonstrates strong experimental improvements, and that our experimental setup is both sound and well-motivated. The reviewer's suggestions regarding experiments on resource usage, prototype-based retrieval, and real-time environmental changes have been very helpful. We have addressed all of the reviewer's comments in our author response and will incorporate these suggestions into the final version. If you have any further questions or comments, please feel free to ask. Thank you once again!

---

### Official Review · Reviewer_zfbw · 2025-03-14

**Overall Recommendation:** 3

**Summary:**

This paper presents World Model Implanting (WorMI), a framework to improve the test-time adaptation of embodied AI agents. This work assumes access to a set of world models which are pre-trained on a set of datasets. During the adaptation phases, they select a subset of models that are most relevant to the input state (done via prototype-based retrieval), and then they perform "world-wise" hierarchical cross attention that is the input to the reasoning model.

**Claims And Evidence:**

Yes. All the claims have experiments to support them.

**Essential References Not Discussed:**

Not to the best of my knowledge.

**Experimental Designs Or Analyses:**

Yes, I have checked the soundness and validity of the experiments. No critical issues in experimentation.

**Methods And Evaluation Criteria:**

Yes.

**Other Comments Or Suggestions:**

No specific typo that I could find.

**Other Strengths And Weaknesses:**

**Strengths**:

1. The paper shows really strong few-shot results in unseen environments as well as on unseen tasks in Alfred and VirtualHome embodied environments.

**Weaknesses**:

1. I find the notion of the world model in the paper to be misleading as the term is predominantly used to mean a transition dynamics model with an optional reward model. However, there are no details regarding the World model training or its description apart from one small paragraph at the beginning of Sec 3.2.

2. The writing of the paper needs to be improved. Lots of missing definitions and missing details (mentioned above as well as in other comments) make me question the reproducibility of this work.

**Questions For Authors:**

1. How are the "world models" trained?

2. What is $I$ in $D_j = {(I, s_t, a_t, s_{t+1})}$ denote?

3. Is it possible to compare the inference time for WorMI and the baselines? I'm curious to see how much time does the selection of relevant $K$ models takes.

4. For figure 5(a) what are WM {1-6} trained on?

5. I am curious if WorMI can adapt to finding objects that are not typically in the "desired" location. For instance for the task of "Place breadslice in microwave," the agent's attention is over the kitchen as it is the most likely place to be -- however if the microwave (for whatever) reason is in the living room/bedroom - does the agent have the capability to explore?

6. What object detection module $\phi_D$ is used? If it's a learned model -- how many of the errors in the final performance are due to either misrecognition or missing the recognitions?

7. [Clarification question]: For the results in Table 3 (a), are the models still trained with World-wise compound attention and only differ in the number of models used to perform the retrieval?

**Relation To Broader Scientific Literature:**

This work is very much related to the field of Embodied AI, and agents being able to adapt in their environment is a very critical problem to address since they are bound to fail to some new unseen environments/tasks.

**Theoretical Claims:**

No significant theory (mostly an empirical work). One exception is the bound for prototype-based similarity that is discussed in the appendix. I have not given much importance to that part of the paper.

---

> ### Author Rebuttal · Authors · 2025-04-01
>
> We sincerely appreciate your detailed comments. We will include the following experimental results and clarifications in the final version.
> > Q1. How are the "world models" trained?
>
> Following the reviewer’s comment, we clarify that the world models imitate the environment by capturing dynamics and affordances.
> Each world model is fine-tuned with LLaMA-3.2-1B using text-based states $s_t$, actions $a_t$, and instructions $I$. Specifically, we use three auxiliary tasks: (1) predict $s_{t+1}$ from $(s_t, a_t)$ to learn transition dynamics; (2) identify feasible actions $a_t$ from $s_t$ to capture affordances; (3) predict $a_t$ from $(s_t, I)$ to account for instruction conditioning.
> The dataset is collected only from seen scenes, and instructions are sampled solely from seen tasks. Training prompts for each world model build on the environment prompts in the appendix, with added instructions for each auxiliary task.
>
> > Q2. What is $I$ in $D_j=(I,s_t,a_t,s_{t+1})$ denote?
>
> $I$ is the embodied task instructions. We will add the descriptions in the main text Line 146.
>
> > Q3. Comparison for the inference time and memory efficiency.
>
> The table below shows the inference times and memory usages among the baselines and our WorMI.
> We use LLaMA-3.2-11B for reasoning and LLaMA-3.2-1B for world models, with the same configurations for all baselines. LLM-Planner’s long in-context prompts increase inference time, and SayCanPay’s use of three models with repetitive action log probability computations further slows performance.
> WorMI uses smaller world models and selects only the K world models, with each domain-specific world model kept much smaller than the reasoning model, easing scalability.
>
> |Model|Inference Time|Memory|
> |-|-|-|
> |LLM+FT, ZSP|298ms|21877MiB|
> |LLM-Planner|401ms|21877MiB|
> |SayCanPay|609ms|46230MiB|
> |WorMI(K=2,N=4)|339ms|30020MiB|
> |WorMI(K=2,N=6)|348ms|33445MiB|
> |WorMI(K=3,N=6)|385ms|33445MiB|
>
> > Q4. For figure 5(a) what are WM {1-6} trained on?
>
> WM1 to WM6 represent six pre-trained world models, with each model trained on a dataset from a distinct domain as demonstrated in Figure 5(b) showing Domain 1 through Domain 6.
>
> > Q5. Exploration capability for WorMI
>
> WorMI is capable of exploration even if the object is located outside its seen domain.
> For instance, if the TV is usually in the living room but is found instead in the bedroom or kitchen, it checks that likely location, and if the TV is missing there, it searches other rooms.
> This works because, beyond each world model’s domain-specific knowledge, the world-to-reasoning alignment also leverages the reasoning model's general knowledge, letting the agent systematically explore instead of relying exclusively on its seen domains.
>
> > Q6. Clarification for object detection module $\phi_D$.
>
> We use the environment’s built-in object detection, which is error-free.
> Therefore, errors due to missed or incorrect object recognition do not affect our method or any of the baselines.
> This setup aligns with text-based simulation environments such as VirtualHome and ALFWorld, where object descriptions are provided without recognition errors, and is consistently applied in other comparison studies as well [1, 2, 3, 4, 5].
>
> In addition, for real-world scenarios requiring visual input, our framework can integrate an object detection module $\phi_D$ by using a vision-language model as its reasoning component, extending the approach beyond purely text-based domains.
>
> The table below shows the performance of multi-modal WorMI, which employs a VLM as its reasoning model, using both text and image states in VirtualHome.
> Multi-modal WorMI exhibits only a slight performance drop compared to WorMI, demonstrating the applicability for  multi-modal experiment setups.
> Additionally, there is certainly room for improvement of Multi-modal WorMI, as we do not have enough time to optimize the hyperparameters.
> Even so, our approach still demonstrates superior performance compared to the baselines.
>
> |Model|SR (↑)|PS (↓)|
> |-|-|-|
> |Multi-modal WorMI|57.65%|17.21|
> |WorMI|66.12%|15.17|
>
> [1] Huang et al. "Language models as zero-shot planners: Extracting actionable knowledge for embodied agents." ICML 2022.
>
> [2] Song et al. "Llm-planner: Few-shot grounded planning for embodied agents with large language models." ICCV 2023.
>
> [3] Hazra et al. "Saycanpay: Heuristic planning with large language models using learnable domain knowledge." AAAI 2024.
>
> [4] Singh  et al. "Progprompt: Generating situated robot task plans using large language models.", ICRA 2023.
>
> [5] Yoo et al. "Exploratory retrieval-augmented planning for continual embodied instruction following.", NeurIPS 2024.
>
> > Q7. Clarification for the configuration in Table 3(a)
>
> Yes. All variants in Table 3(a) use the same world-wise compound attention during training. They only differ in which world models are selected at test time. WorMI-E uses all models, while WorMI-R randomly picks a subset, and WorMI does prototype-based retrieval.

---

### Official Review · Reviewer_MES3 · 2025-03-17

**Overall Recommendation:** 3

**Summary:**

This paper presents WorMI, a framework enabling embodied agents to adapt to new domains at test time by combining large language models with domain-specific world models. It introduces prototype-based world model retrieval and world-wise compound attention to effectively integrate knowledge from multiple models. Experiments show WorMI outperforms existing methods in zero-shot and few-shot scenarios, demonstrating robust adaptation to unseen domains. The framework's design allows for scalable, efficient deployment in real-world settings where adaptability and data efficiency are crucial.

**Claims And Evidence:**

Yes

**Essential References Not Discussed:**

No

**Experimental Designs Or Analyses:**

Yes

**Methods And Evaluation Criteria:**

Yes

**Other Comments Or Suggestions:**

See the weaknesses

**Other Strengths And Weaknesses:**

Strengths:
1) The paper presents a novel framework (WorMI) that creatively combines prototype-based retrieval with a world-wise compound attention mechanism for embodied AI adaptation. This approach addresses a significant challenge in the field by enabling dynamic composition of world models at test time, representing a meaningful advancement over previous methods.
2)The work tackles a crucial problem for real-world embodied AI applications—adaptation to new domains without extensive retraining. The demonstrated performance improvements, particularly in zero-shot and few-shot scenarios, suggest practical impact and scalability.
3)The paper is well-structured.


Weaknesses:
1)While the combination of methods is novel, the individual components (prototype-based retrieval, attention mechanisms, meta-learning) are not entirely new. Some readers might argue that the innovation is incremental rather than groundbreaking.
2)The framework's performance is inherently dependent on the underlying language model, which could be seen as a limitation since it inherits any weaknesses of the LLM.
3) What specific challenges have you identified in deploying WorMI in real robotic systems with sensorimotor embodiments?

**Questions For Authors:**

See the weaknesses

**Relation To Broader Scientific Literature:**

The paper introduces WorMI, a framework that advances embodied AI by enabling test-time adaptation through dynamic composition of domain-specific world models with LLMs, building upon and extending prior work in model composition, knowledge retrieval, and meta-learning. The novel prototype-based retrieval and compound attention mechanisms in WorMI efficiently select and integrate relevant models, addressing limitations in computational efficiency and adaptability found in previous approaches, and demonstrate superior performance in zero-shot and few-shot scenarios across established benchmarks.

**Theoretical Claims:**

Yes

---

> ### Author Rebuttal · Authors · 2025-04-01
>
> We sincerely appreciate your detailed comments. We will include the following experimental results and clarifications in the final version.
>
> > Q1. While the combination of methods is novel, the individual components (prototype-based retrieval, attention mechanisms, meta-learning) are not entirely new. Some readers might argue that the innovation is incremental rather than groundbreaking.
>
> To the best of our knowledge, our model implanting framework is the first to enable selective addition and removal of domain-specific world models in an agent’s policy at test time.
> The novelty lies in how we orchestrate them to enable a plug-and-play composition of world models.
> Rather than training a single monolithic model or simply combining multiple models externally, we propose a framework where each world model is fully implanted in the reasoning model only when relevant to the current domain.
>
> Our approach is novel in its three-fold strategy. First, prototype-based retrieval efficiently selects only the most relevant world models for an unseen domain, reducing overhead.
> Second, our compound attention mechanism fuses domain-specific knowledge and aligns it with LLM reasoning at test time. Finally, meta-learning enables seamless adaptation to new models or domains without retraining components.
> By combining these three ideas, we achieve a flexible test-time architecture that is easily extensible and maintainable, providing a clear step forward over existing methods that rely solely on in-context learning or static model ensembles.
>
> > Q2. The framework's performance is inherently dependent on the underlying language model, which could be seen as a limitation since it inherits any weaknesses of the LLM.
>
> WorMI leverages domain-specific world models to reduce hallucination and bias instead of relying solely on the LLM’s own knowledge. As shown in Table 4, it remains robust even with a smaller LLM while keeping expansion costs low. However, completely eliminating LLM weaknesses remains challenging. These limitations are explicitly addressed in the main text.
>
> > Q3. What specific challenges have you identified in deploying WorMI in real robotic systems with sensorimotor embodiments?
>
> We consider multi-modality is critical for real robotic systems, which often rely on various sensor inputs. Our WorMI framework is designed to support this by allowing the reasoning model and individual world models to come from different modalities.
>
> The table below shows the performance of multi-modal WorMI, which employs a VLM as its reasoning model, using both text and image states in VirtualHome.
> Multi-modal WorMI exhibits only a slight performance drop compared to WorMI, demonstrating the applicability for multi-modal experiment setups.
> Additionally, there is certainly room for improvement of Multi-modal WorMI, as we do not have enough time to optimize the hyperparameters.
> Even so, our approach still demonstrates superior performance compared to the baselines.
> We will include these experimental results in the final version.
>
> | Model             | SR (↑) | PS (↓) |
> |------------------|--------|--------|
> | Multi-modal WorMI| 57.65% | 17.21  |
> | WorMI            | 66.12% | 15.17  |

---

### Decision · Program_Chairs · 2025-05-01

**Decision:**

Accept (poster)

**Comment:**

In this work, the authors propose WorMI, a world model implanting framework that can leverage a set of pre-trained domain-specific (neural) world models during test-time by fusing their representations with reasoning LLMs. The main contributions include a prototype-based retrieval mechanism, which selects relevant world models from a set of candidate based on the specific test domain; a compound attention mechanism that integrates the selected world models with the reasoning model during test-time.

All reviewers agree that the WorMI framework is novel; the idea of composing pretrianed world models and fusing with reasoning LLMs is interesting; the prototype-based retrieval mechanism and the compound attention mechanism make sense and are well designed.

The reviewers' concerns were mostly regarding the insufficient experiments and analysis. The authors did a good job addressing the concerns by providing a set of additional experiments, new results and some new qualitative analysis have indeed strengthened the work.

---

Additional comments from AC:

1. I agree with Reviewer sgRi that the work could benefit from more insights/discussions on leveraging (highly) out-of-distribution world models, for example, can agents use a set of VirtualHome world models in a ALFWorld environment, or can agents use a set of world models pretrained on a wide spectrum of embodied tasks in an unseen benchmark? I acknowledge that there are already "ood" test sets in benchmarks like ALFWorld, but I believe a more realistic setting would be trying to leverage pre-trained world models on completely unseen scenarios.
2. Related to the above point, it may be worth investigating how the diversity of pre-trained world models affects WorMI's performance, especially on ood settings. I believe to fully unleash the potentials of test-time world model fusing, one might need to have a large and at the same time very diverse set of world models.

---

AC Recommendation:

The initial scores were 3/2/3/2 (avg 2.5); after the rebuttal, one reviewer increased their score, resulting 3/2/3/3 (avg 2.75). It's worth noting that the reviewer who gave the remaining score of 2 only acknowledged but did not respond to the authors, which may suggest they do not think the rebuttal is convincing enough.
To me this is a good example where the reviewers helped the authors to improve their work by suggesting reasonable additional experiments, the authors also did well on addressing these requests.